# TREE-AS-A-PROMPT: BOOSTING BLACK-BOX LARGE LANGUAGE MODELS ON FEW-SHOT CLASSIFICATION OF TABULAR DATA

## ABSTRACT

Large Language Models (LLMs) have achieved remarkable success across various natural language processing tasks, yet their application to tabular data presents unique challenges in terms of performance and interpretability. The intrinsic structure and characteristics of tabular data necessitate innovative strategies to unlock the full potential of LLMs in this domain. Recognizing the proficiency of decision trees in handling tabular data, we introduce Tree-as-a-Prompt in this paper. In addition to the original query, we propose to convert a decision tree into prompts and feed them into the LLM, aiming to enhance the performance of LLMs on tabular data. The decision tree is treated as a part of the composite model alongside the LLM and is optimized based on the LLM's predictions. Our results demonstrate that appending the decision tree as a prompt boosts the performance of LLMs on tabular data significantly. Additionally, the decision tree serves as an instrumental tool in elucidating the predictions of LLMs, thereby enhancing the model interpretability for different applications.

## 1 INTRODUCTION

Large Language Models (LLMs) (OpenAI, 2023; Touvron et al., 2023; Narang & Chowdhery, 2022; Sun et al., 2021) have revolutionized the field of Natural Language Processing (NLP), demonstrating superior proficiency in a variety of tasks including text completion, translation, and sentiment analysis. These models leverage vast amounts of data to comprehend and generate human-like text, showcasing the potential of deep learning in understanding natural language. However, despite their remarkable success in NLP tasks, LLMs exhibit limitations when applied to tabular data. For example, given a health record, LLM may not be able to predict the risk of diseases with a reasonable explanation. The inherent structure and diversity of tabular data pose challenges in performance and interpretability, limiting the effectiveness of LLMs in fully exploiting such data.

Several strategies exist for enhancing the efficacy of LLMs on tabular data. One prevalent method is fine-tuning the model on tabular datasets, with works like TabLLM (Zha et al., 2023) illustrating the promise of fine-tuned LLMs for few-shot classification on such data. However, these approaches often incur significant computational costs and necessitate access to model parameters. Another strategy is prompt tuning (Lester et al., 2021; Liu et al., 2022; Qin & Eisner, 2021), which adjusts the tokens prepended to the input text while keeping the LLM fixed. This method, however, typically requires access to the embedding space and output likelihood, rendering it inapplicable to black-box LLMs such as GPT-3.5 and GPT-4 (OpenAI, 2023). In this paper, our goal is to enhance the classification capabilities of LLMs on tabular data in a black-box setting. On the other hand, when an individual queries the black-box LLM, the user usually has a limited amount of data. One scenario is that individuals use LLMs for preliminary disease diagnosis by inputting only a few examples (Oniani et al., 2023). Thus, we focus on the few-shot classification of tabular data.

While LLMs have attracted a lot of attention, decision tree (Myles et al., 2004) is a classic machine learning model and has long been recognized for the efficacy in handling tabular data when compared with deep learning models (McElfresh et al., 2023). In a decision tree, input data is classified by splitting rules. The hierarchical structure and rule-based splitting criteria allow it to effectively capture the relationships and patterns within tabular datasets. Furthermore, the transparency of their

decision-making process makes them a valuable tool in applications where model interpretability is crucial such as finance and healthcare, providing insights into the underlying mechanisms of the model and facilitating trust in model predictions.

Given the proficiency of decision trees in classifying tabular data, a pertinent question arises: can we leverage decision trees to enhance LLMs? Essentially, a decision tree is a combination of split rules, which can be easily converted to texts and fed into LLMs. In this paper, we propose a novel approach, Tree-as-a-Prompt (TAP), to harness the strengths of decision trees in enhancing the performance and interpretability of LLMs on tabular data. By converting a decision tree into prompts and feeding them, along with the original query, into the LLM, we aim to create a synergistic model that combines the complementary strengths of both components. The decision tree is optimized based on the LLM's predictions, establishing a collaborative relationship between the interpretability of the tree and the learning capacity of the LLM. Our approach can also be extended to enable tuning of LLMs in the federated learning setting (Kairouz et al., 2021; Yang et al., 2019), where multiple parties use our approach to train the tree, and the trees are merged to enable knowledge aggregation.

Our main contributions are summarized as follow:

- We show that appending trees as prompts to the original query is a simple and effective strategy to enhance the capabilities of LLMs on tabular data.
- We introduce the concept of Tree-in-the-Loop, a methodology that trains a decision tree utilizing the feedback from LLMs, enabling it to effectively encapsulate the knowledge necessary to aid the LLMs' predictions.
- Through comprehensive experiments on multiple applications, we validate that our approach outperforms the exclusive use of either LLMs or trees on few-shot classification of tabular data, and we illustrate that the generated trees can serve as explanatory criteria to elucidate the decision-making behavior of LLMs.

## 2 BACKGROUND AND RELATED WORK

### 2.1 LLMS FOR TABULAR DATA

While LLMs have been extensively applied to natural language tasks, their exploitation for tabular data remains relatively underexplored. Existing studies (Wang et al., 2023; Herzig et al., 2020; Zhang et al., 2023; Hegselmann et al., 2023) primarily concentrate on training or tuning strategies aimed at enhancing LLM performance on tabular data. One related study in this context is TabLLM (Hegselmann et al., 2023), which employs fine-tuning techniques to augment the LLM's capability in few-shot classification on tabular data. The study delves into various serialization techniques to convert tabular data into text, demonstrating that fine-tuning the T0 model (Sanh et al., 2021) on a limited set of tabular examples can, in certain scenarios, outperform tree-based methods. In contrast to these model-updating approaches, our objective is to augment the predictive prowess of existing black-box LLMs on tabular data by strategically modifying the input prompts.

### 2.2 PROMPT TUNING

Prompt tuning (Lester et al., 2021; Liu et al., 2022; Zhong et al., 2021; Li & Liang, 2021; Qin & Eisner, 2021; Liu et al., 2023) is a parameter-efficient way to boost LLMs on a targeted task. The basic idea of prompt tuning is to optimize the prefix prompts while freezing the LLMs. For example, Li & Liang (2021) optimizes the log-likelihood objective on the input data given the trainable prefix matrix and the frozen language model. Lester et al. (2021) prepend the input query with special tokens and tune the embeddings of these tokens directly, which adds fewer parameters to tune. Although prompt tuning does not require updating the LLM, they still need access to the output likelihood and the intermediate embedding layers to tune and prepend the embedding of prompts. Thus, they are not applicable in the black-box LLM setting such as GPT-4 (OpenAI, 2023).

### 2.3 DECISION TREES

Decision Trees (Loh, 2011; Maimon & Rokach, 2014; Quinlan, 1986; 2014) are classic and fundamental models used in machine learning and data mining. The core concept behind decision

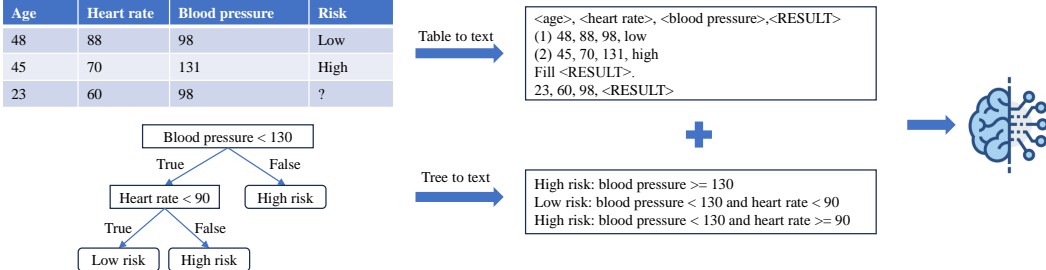

Figure 1: An example of appending trees to the prompts. Besides the original query, we convert the tree into text by describing the split rules and feed them into the LLM.

trees is to represent rules in a tree-like structure, where internal nodes denote tests on an attribute, branches represent the outcome of the test, and leaf nodes hold the predictions. The construction of a decision tree involves recursively partitioning the dataset into subsets based on the attribute that results in the maximum information gain or the smallest impurity, such as Gini impurity (Loh, 2011). Decision trees are easy to understand, interpret, and visualize, which makes them highly attractive for applications where interpretability is crucial.

## 2.4 PROMPT ENGINEERING

Various studies (Wei et al., 2022; Qin & Eisner, 2021; Yao et al., 2023) have delved into exploring techniques to prompt LLMs, aiming to enhance their problem-solving capabilities. For instance, Chain-of-thought (Wei et al., 2022) introduces an innovative method of incorporating a "chain of thoughts" to bridge the input and answer, rather than feeding them directly to LLMs. These chains represent meaningful intermediate steps that can bolster the reasoning abilities of LLMs. Another notable methodology, Tree-of-thoughts (Yao et al., 2023), facilitates multiple reasoning paths over thoughts, thereby generating more refined results. Our approach draws parallels with these studies, but with a distinctive focus. These approaches do not investigate the application on the classification of tabular data. Instead of manually crafting prompts to guide thought processes for tasks such as solving mathematical word problems, we devise a novel method to train a decision tree. This tree captures the knowledge embedded in tabular data and conveys this knowledge to LLMs, thereby enhancing their predictive accuracy on tabular datasets.

## 3 TREES CAN IMPROVE LLMs

We commence our investigation by examining whether utilizing pre-trained decision trees can enhance the performance of LLMs in the few-shot classification of tabular data. The overarching structure for appending trees is depicted in Figure 1. In this methodology, both the tabular data and the decision tree are converted into text and subsequently fed into the LLM. This facilitates the prediction of an instance based on the provided features.

**Table to Text** We explore three distinct templates for converting tables into text, as illustrated in Figure 2: the text template, the list template, and the tabular template. (1) Text Template: The tabular data is articulated as "The *{feature name}* is *{feature value}*". (2) List Template: The tabular data is represented as a list of "*{feature name}*: *{feature value}*". (3) Table Template: The feature names are initially introduced, followed by the presentation of the feature values. The text and list templates, introduced in TabLLM (Hegselmann et al., 2023), have been demonstrated to outperform other table serialization approaches such as those employing LLMs for text generation. The table template is a new approach proposed by us. We evaluate the three templates in Figure 3a utilizing GPT-3.5 in a few-shot classification setting (#shots=16) on a *diabetes* tabular dataset. The results indicate that our proposed method, the table template, yields the most favorable outcome. Furthermore, the table template requires fewer tokens compared to the other two templates, as it eliminates the need for repetitive feature name articulation, thereby reducing computational costs.

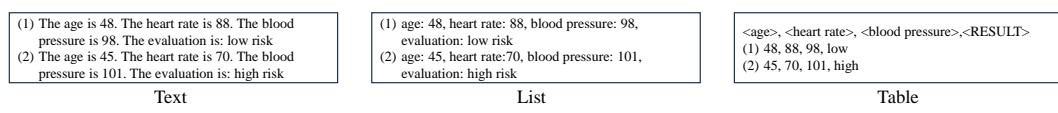

Figure 2: Three templates to convert tabular data into texts.

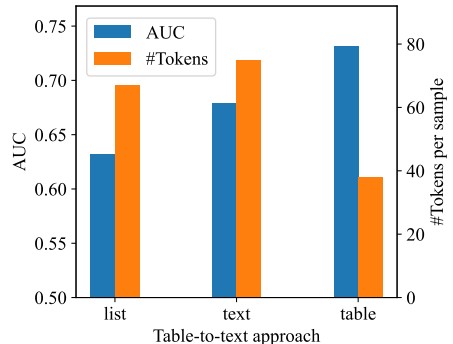

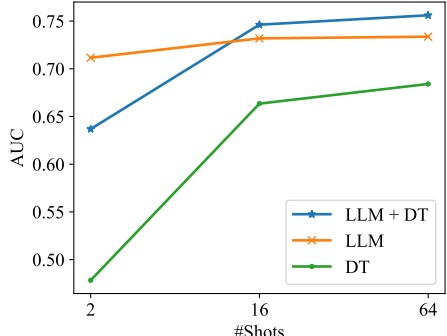

(a) Comparison between different table to text approaches.

(b) The performance of appending trees as prompts. We run five trials and report the mean AUC.

Figure 3: Few-shot classification of GPT-3.5 on *diabetes*.

**Tree to Text**  Converting a tree to text is a rather intuitive process. As depicted in Figure 1, each path from the root node to a leaf node is described using text in the format "*{label}: {split rule}*". These formulated rules are then provided to the LLM to assist in enhancing its predictive capabilities.

**Feeding Pre-trained Trees into LLMs**  We initiate our exploration by utilizing the *diabetes* dataset to illustrate that simply incorporating a pre-trained tree into the prompts can enhance the predictive accuracy of LLMs. Specifically, we commence by training a CART classification tree (Loh, 2011) utilizing the available training data. Subsequently, both the training data and the derived tree are converted into prompts and introduced to GPT-3.5. Following this, we transform the test data into the query prompt and assess the predictive performance of GPT-3.5. The resulting predictions, obtained by using only the LLM, solely employing the decision tree (DT), and incorporating the tree into the LLM input (LLM+DT), are illustrated in Figure 3b. For LLM, we conduct few-shot in-context learning, i.e., feeding the instances by prompting. For DT, we use the instances to train the tree model. For LLM+DT, in addition to in-context examples, we also provide rules from the pretrained DT. It is observed that as the number of training instances increases and the tree's accuracy improves, the performance of the LLMs is enhanced through the addition of trees to the input, indicating the LLM's successful interpretation of prompts derived from the decision tree. Thus, it is promising to append trees to the query to enhance LLMs. For the experiments on more datasets, please refer to Section 5.2.

## 4 TREE-IN-THE-LOOP

In Section 3, we demonstrate that the performance of LLMs can be augmented by incorporating a pre-trained tree as prompts. However, in scenarios where the number of training examples is significantly limited, a pre-trained tree may not offer any utility, resulting in LLM+DT underperforming the LLM alone (e.g., the experiments in Figure 3b when #shots=2). LLMs, being inherently knowledgeable, can often make superior predictions compared to trees, especially when they possess background knowledge pertinent to the task. In such cases, the introduction of tree prompts can potentially mislead the LLM. To address this and effectively harness the knowledge inherent in LLMs, we propose the tree-in-the-loop methodology. This approach, illustrated in Figure 4, involves utilizing the feedback generated by the LLM to train the decision tree.

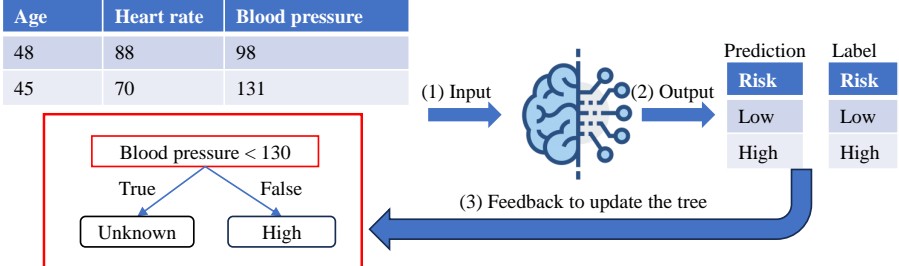

Figure 4: The general pipeline of updating the decision tree from the feedback of the LLM. There are three steps in each round: 1) Input the query prompts and tree prompts to the LLM; 2) Compute the loss based on the prediction of the LLM (and the tree); 3) Update a tree node with the minimal loss.

## 4.1 DECISION TREE WITH UNKNOWN LEAVES

In traditional decision tree models, each leaf node houses a prediction value, ensuring that every potential input instance receives a prediction. However, in our methodology, where decision trees are transformed into text, assigning a definite prediction value to every leaf node is not necessary, especially for nodes without high confidence. To accommodate instances where the tree is unable to render confident predictions based on the split rules, we introduce an "unknown" option. During the conversion of the tree into text, only the paths from the root node to the leaf nodes that are not designated as "unknown" are serialized. The incorporation of unknown leaves allows us to filter out noisy rules and retain only those that are valuable for both prediction and explanation.

## 4.2 GREEDY TREE TRAINING WITH LLM FEEDBACK

Given the substantial computational cost associated with traversing all possible tree structures to identify the optimal one, we adopt a greedy strategy to update the tree, progressing from the top to the bottom. At each node, we explore all potential split values to pinpoint the one that yields the minimal loss, designating it as the optimal node. Specifically, during the update of a node, we consider all possible leaf values as the child nodes of the current node. In the context of binary classification, the leaf value can be 0, 1, or "unknown". Subsequently, for each tree structure generated, we convert it into prompts and append them to the original query prompt, thereby soliciting predictions from the LLM. The loss is calculated based on the $L_1$ norm distance between the predicted values and the actual labels, expressed as $\ell_{LLM} = \frac{1}{|\mathcal{D}|} \sum_i |\mathbf{y}_i^p - \mathbf{y}_i|$, where $\mathcal{D}$ represents the training data, $\mathbf{y}^p$ denotes the values predicted by the LLM, and $\mathbf{y}$ signifies the actual labels.

To maintain the interpretability of the tree and avoid the potential misguidance of the LLM by the tree rules, we incorporate a regularization term into the objective. This term evaluates the predictions made solely by the tree. Given that the predictions from tree leaves might be unknown, we introduce a hyperparameter $\mu$ to represent the loss associated with unknown predictions. Suppose $\mathbf{y}^{pt}$ is the prediction of the training data from the tree, the loss attributed to the tree is

$$\ell_{tree} = \frac{1}{|\mathcal{D}|} \sum_i \begin{cases} |\mathbf{y}_i^{pt} - \mathbf{y}_i| & \text{if } \mathbf{y}_i^{pt} \text{ is not } unknown \\ \mu & \text{if } \mathbf{y}_i^{pt} \text{ is } unknown \end{cases}. \tag{1}$$

Then, our final loss to evaluate a tree structure is $\ell = \ell_{LLM} + \lambda \ell_{tree}$, where $\lambda$ is a hyperparameter to control the weight of the regularization term.

The comprehensive algorithm is presented in Algorithm 1. For clarity in presentation, we employ $\mathbf{T}$ to denote the tree model, wherein $\mathbf{T}[i]$ signifies the $i$-th tree node of $\mathbf{T}$ in a top-to-bottom sequence (for instance, $\mathbf{T}[1]$ is the root node). Initially, we generate the query prompts from the training data utilizing the table template, as introduced in Section 3 (Line 1). Additionally, we formulate candidate split values derived from the feature values, which are generated based on histograms. Specifically, these candidate split values uniformly divide the training data into distinct bins (Ke et al., 2017) (Line 2). For each tree node, we explore all feasible split values and leaf values and compute the loss according to equation 1 (Line 3-17). If the calculated loss is found to be smaller than the current minimal loss, the tree is updated with the current nodes (Line 18-21). The tree undergoes updates until it attains the maximum depth or until the loss reaches a minimal state (Line 22-24).

---

**Algorithm 1:** The TAP algorithm with tree-in-the-loop.

---

**Input:** Depth of tree $E$. Training data $\mathcal{D} = \{\mathbf{x}, \mathbf{y}\}$
**Output:** The final tree model $\mathbf{T}$

---

1   QueryPrompts $\leftarrow Table2Text(\mathcal{D})$ // Generate query prompts
2   $\mathbf{S} \leftarrow ProposeSplitPoints(\mathbf{x})$ // Generate split candidates
3   /* Update tree node one by one */
4   **for** $i = 1, ..., 2^E - 1$ **do**
5      $\ell_{min} \leftarrow \infty$
6      /* Traverse all possible split candidates */
7      **for** $s$ in $\mathbf{S}$ **do**
8          $\mathbf{T}' \leftarrow \mathbf{T}$
9          $\mathbf{T}'[i] \leftarrow s$ // set current node to candidate value
10         /* Traverse all possible leaf values */
11         **for** (LeftValue, RightValue) in $\{0, 1, \text{unknown}\} \times \{0, 1, \text{unknown}\}$ **do**
12             $\mathbf{T}'[2i] \leftarrow$ LeftValue // set the left child of node i
13             $\mathbf{T}'[2i + 1] \leftarrow$ RightValue // set the right child of node i
14             TreePrompts $\leftarrow Tree2Text(\mathbf{T}')$
15             $\mathbf{y}^p \leftarrow LLM(\text{QueryPrompts}, \text{TreePrompts})$
16             $\mathbf{y}^{pt} \leftarrow \mathbf{T}'(\mathbf{x})$
17             $\ell \leftarrow \ell_{LLM}(\mathbf{y}^p, \mathbf{y}) + \lambda \ell_{tree}(\mathbf{y}^{pt}, \mathbf{y})$
18             /* Update the node if the loss is minimal */
19             **if** $\ell < \ell_{min}$ **then**
20                $\ell_{min} \leftarrow \ell$
21                $\mathbf{T} \leftarrow \mathbf{T}'$
22             **if** $\ell == 0$ **then**
23                **return** $\mathbf{T}$ // early stopping

24   **return** $\mathbf{T}$

---

## 5   EVALUATION

### 5.1   EXPERIMENTAL SETUP

**Approaches**   We compare the following approaches in the experiments: 1) LLM: we directly feed the training data to the LLM by prompting and then ask it to make prediction on the test data; 2) DT: we use scikit (Pedregosa et al., 2011) to train a CART (Loh, 2011) model on the training data, which is used for prediction on the test data; 3) XGB: we use XGBoost (Chen & Guestrin, 2016) instead of CART to train the decision tree; 4) TAP-DT: we use our proposed tree-as-a-prompt where the tree is the pretrained DT as introduced in Section 3; 5) TAP-OT: we use our proposed tree-as-a-prompt where the tree is trained using our proposed tree-in-the-loop approach in Section 4; 6) OT: After training a tree in TAP-OT, we use the tree solely for inference. For the detailed prompts, please refer to Appendix A. For TAP-OT, we tune $\lambda \in \{0, 0.1, 1, 2, 4\}$ and $\mu \in \{0, 0.2, 0.6, 1, 2\}$. We also try removing the unknown leaves in the experiments to cover the case when $\mu$ is extremely large.

**Datasets and Tasks**   We conduct experiments on four public tabular datasets from UCI machine learning repository (Asuncion & Newman, 2007) covering tasks from disease prediction to age prediction: *diabetes*, *blood*, *car*, and *abalone*. For each dataset, we randomly sample 100 instances as the test dataset and then sample a few instances from the remained data for few-shot learning. For the detailed statistic and task of each dataset, please refer to Appendix B.

**Models**   By default, we use GPT-3.5 in the experiments. Specifically, we use the OpenAI API to access and query the June version of GPT-3.5 (gpt-3.5-turbo-0613). We set the GPT *temperature* parameter to 0 to improve its stability and reproducibility given the same input. For experiments on more LLMs, please refer to Appendix C. For the decision tree, we set the maximum number of depth to 3 for all approaches by default to ensure a fair comparison. The number of bins is set to 10.

Table 1: The AUC of different approaches in the few-shot classification setting. We run five trials for each experiment and report the mean AUC and std. We use ✗ to indicate the case that the tree cannot be successfully trained due to a limited number of instances.

| Datasets | #Shots | LLM | DT | XGB | TAP-DT | OT | TAP-OT |
|---|---|---|---|---|---|---|---|
| diabetes | 1 | **0.71**±0.04 | ✗ | ✗ | ✗ | 0.50±0.06 | 0.70±0.07 |
| | 2 | 0.71±0.06 | 0.48±0.06 | ✗ | 0.64±0.07 | 0.52±0.04 | **0.73**±0.02 |
| | 4 | 0.68±0.04 | 0.50±0.07 | ✗ | 0.69±0.04 | 0.58±0.04 | **0.72**±0.03 |
| | 8 | 0.67±0.05 | 0.63±0.07 | 0.59±0.11 | 0.68±0.04 | 0.65±0.07 | **0.70**±0.06 |
| | 16 | 0.73±0.05 | 0.66±0.07 | 0.67±0.09 | 0.75±0.06 | 0.67±0.06 | **0.77**±0.03 |
| | 32 | **0.75**±0.02 | 0.69±0.05 | 0.67±0.06 | 0.74±0.04 | 0.68±0.04 | **0.75**±0.02 |
| car | 1 | 0.72±0.05 | ✗ | ✗ | ✗ | 0.56±0.10 | **0.83**±0.04 |
| | 2 | 0.78±0.07 | 0.64±0.18 | ✗ | **0.81**±0.08 | 0.63±0.08 | 0.78±0.04 |
| | 4 | 0.78±0.05 | 0.68±0.10 | ✗ | 0.77±0.06 | 0.62±0.08 | **0.80**±0.04 |
| | 8 | 0.89±0.03 | 0.85±0.13 | 0.67±0.14 | 0.86±0.06 | 0.71±0.12 | **0.91**±0.01 |
| | 16 | 0.92±0.03 | 0.85±0.13 | 0.83±0.06 | 0.90±0.03 | **0.94**±0.04 | 0.91±0.04 |
| | 32 | 0.94±0.02 | **0.95**±0.03 | 0.77±0.13 | 0.91±0.03 | **0.95**±0.04 | 0.93±0.01 |
| blood | 1 | 0.59±0.05 | ✗ | ✗ | ✗ | 0.52±0.05 | **0.67**±0.06 |
| | 2 | 0.68±0.01 | 0.69±0.05 | ✗ | 0.69±0.02 | 0.67±0.06 | **0.70**±0.04 |
| | 4 | **0.62**±0.09 | 0.54±0.18 | ✗ | 0.60±0.09 | 0.61±0.09 | **0.62**±0.07 |
| | 8 | 0.65±0.05 | 0.62±0.08 | **0.66**±0.09 | 0.65±0.07 | 0.65±0.09 | 0.65±0.07 |
| | 16 | 0.62±0.06 | **0.63**±0.08 | **0.63**±0.09 | **0.63**±0.04 | 0.51±0.08 | 0.62±0.07 |
| | 32 | **0.67**±0.03 | 0.66±0.09 | 0.65±0.07 | 0.64±0.04 | 0.65±0.06 | **0.67**±0.03 |
| abalone | 1 | **0.71**±0.04 | ✗ | ✗ | ✗ | 0.50±0.08 | 0.69±0.06 |
| | 2 | **0.71**±0.02 | 0.60±0.20 | ✗ | 0.70±0.05 | 0.56±0.20 | 0.69±0.05 |
| | 4 | **0.72**±0.03 | 0.57±0.24 | ✗ | **0.72**±0.02 | 0.59±0.13 | 0.71±0.02 |
| | 8 | 0.71±0.03 | 0.59±0.12 | 0.64±0.11 | 0.74±0.01 | 0.63±0.17 | **0.75**±0.02 |
| | 16 | 0.73±0.02 | 0.68±0.07 | 0.69±0.09 | **0.75**±0.02 | 0.56±0.10 | **0.75**±0.03 |
| | 32 | 0.72±0.02 | 0.70±0.07 | 0.72±0.07 | 0.73±0.04 | **0.76**±0.05 | 0.71±0.03 |

**Setup**  We run the experiments on a Linux server with 4x AMD EPYC 7543 32-Core Processor. Since we use OpenAI API to query GPT, there is no strict requirement on the capacity of the machine. By default, we run five trials for each experiment and report the mean and standard derivation.

## 5.2  Model Performance

We evaluate the efficacy of our proposed methods against baseline approaches in a few-shot classification task, varying the number of training instances from 1 to 32. The results are detailed in Table 1. Several key observations can be drawn from these results: 1) *Effectiveness of TAP-OT*: TAP-OT outperforms other methods, securing the best results in a majority of scenarios and underscoring the effectiveness of our approach. In instances where baseline methods achieve superior performance, TAP-OT still remains closely competitive, registering near-optimal accuracy. 2) *Superiority of OT Over DT*: Our tree generally exhibits better performance than DT. While DT is trained solely on limited training instances, our tree amalgamates knowledge from both the training data and the LLM. This fusion results in the formulation of more effective rules for classification tasks. 3) *LLM Performance in Few-Shot Learning*: Notably, in a few-shot learning environment, DT tends to underperform

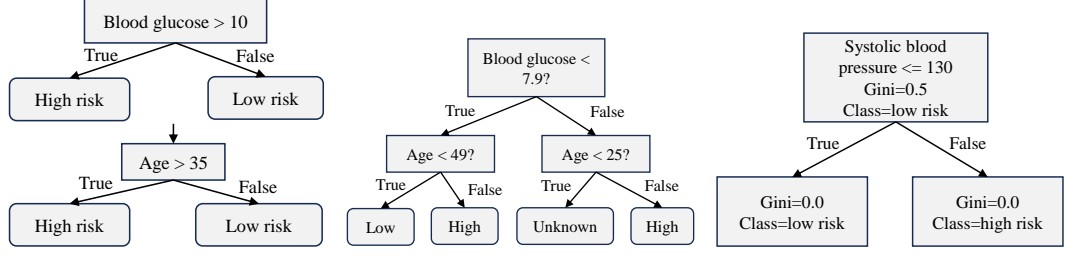

(a) An example of the unstructured tree generated by the LLM.

(b) Visualization of a tree trained by TAP-OT.

(c) Visualization of a tree in DT.

Figure 6: Comparison of interpretability of different models.

compared to LLM, particularly when the availability of training instances is scarce. This observation corroborates the inherent capacity of LLMs to leverage background knowledge for specific tasks (e.g., disease prediction), suggesting their promising applicability over trees in data-limited contexts.

## 5.3 INTERPRETABILITY

We compare the interpretability of three different models, LLM, the tree trained using our approach, and the pretrained CART in Figure 5 and Figure 6 using an example of risk intensity level prediction. Figure 5 shows the example of the prediction and explanation of the LLM. LLMs are knowledgeable, and thus they can make good predictions on the given tabular data in some cases. However, in providing explanations, the LLM typically enumerates the factors contributing to the prediction, exhibiting several limitations: 1) Normal feature values (e.g., age), which typically are not crucial for reasoning, are also reported; 2) The influential factors are delineated for each individual instance, hindering users from readily identifying systematic, quantified key contributors to the task. 3) No knowledge of the relation between multiple factors is provided.

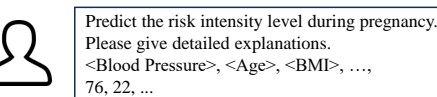

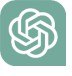

Figure 5: An example of LLM prediction with explanations.

We also ask the LLM to summarize the rules in a tree format, which is presented in Figure 6a. The LLM falls short in generating a structured binary tree; instead, it merely links the trees representing various factors, neglecting to delineate the relationships between them. In contrast, as depicted in Figure 6b, our tree effectively visualizes the task rules and is readily interpretable compared to the explanation offered by the LLM. Additionally, when juxtaposed with the CART shown in Figure 6c, our tree emerges as more rational. The CART model is susceptible to overfitting, simply labeling the instances as low risk given a normal feature value. Our tree introduces the "unknown" value for instances with scarce normal feature values, thereby allowing the LLM to base predictions on additional features not encompassed by the tree.

## 5.4 MULTIPLE TREES

In Algorithm 1, we delineate the process of training a single tree. However, our methodology can also be seamlessly extended to accommodate the training of multiple trees through feature bagging. Specifically, we partition the feature space into several random subsets and independently train a tree for each. Subsequently, the prompts generated by all these trees are fed into the LLM. The outcomes, utilizing three trees (#shots=8), are showcased in Table 2. Evidently, our strategy surpasses both the standalone LLM and the random forest baselines in all four datasets, underscoring the scalability and adaptability of our approach.

Table 2: Comparison between different approaches with multiple trees. RF is a random forest approach provided by scikit.

|  | LLM | RF | TAP-OT |
|---|---|---|---|
| diabetes | 0.67±0.05 | 0.56±0.04 | **0.70**±0.05 |
| car | 0.89±0.03 | 0.71±0.12 | **0.93**±0.02 |
| blood | 0.65±0.05 | 0.65±0.07 | **0.66**±0.07 |
| abalone | 0.71±0.03 | 0.60±0.18 | **0.74**±0.01 |

Table 3: Ablation study by removing the unknown option in leaf values or the regularization term.

|  | TAP-OT | w/o unknown | w/o $\ell_{tree}$ |
|---|---|---|---|
| diabetes | **0.70**±0.06 | 0.69±0.05 | 0.67±0.06 |
| car | **0.91**±0.01 | 0.88±0.01 | 0.83±0.08 |
| blood | **0.65**±0.07 | 0.61±0.06 | 0.64±0.06 |
| abalone | **0.75**±0.02 | 0.74±0.03 | 0.73±0.02 |

## 5.5 ABLATION STUDY

We explore the impact of incorporating unknown leaves and the regularization term, as depicted in equation 1, within our methodology. The corresponding results are showcased in Table 3, wherein the number of training instances is fixed at eight. The outcomes indicate that our approach derives substantial benefits from the introduction of both the unknown option and the tree-based regularization term. The improvement of these techniques is about 2-3% AUC on average.

## 5.6 APPLICATION ON FEDERATED LEARNING

Federated Learning (FL) (Kairouz et al., 2021; Yang et al., 2019; Li et al., 2021) is a popular machine learning paradigm wherein multiple parties collaboratively train a machine learning model without exchanging their local data, offering privacy advantages. Fine-tuning a public LLM with data from various parties within a federated framework presents significant challenges such as expensive computation costs and easily to be overfitted.

Our methodology offers a natural solution for parameter-efficient federated tuning of LLMs on tabular data. Rather than fine-tuning the LLM directly, each participating entity can locally employ our approach to train a decision tree.

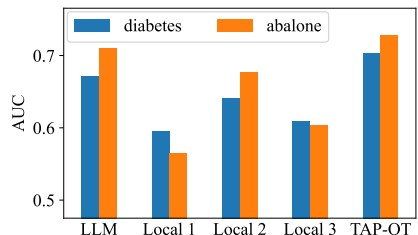

Figure 7: Application of TAP in FL.

Subsequently, these trees can be amalgamated into an ensemble, facilitating the consolidation of knowledge across parties. During inference, the prompts from all trees are appended in our model. To emulate a vertical FL scenario, we partition *diabetes* and *abalone* into three equal, random subsets along the feature dimension. We then compare the efficacy of using only the LLM for predictions, solely relying on locally trained trees, and employing our integrated approach, as illustrated in Figure 7. The results indicate that our method significantly enhances performance by harnessing the collective knowledge of multiple parties.

## 6 CONCLUSION

In this work, we have introduced a novel methodology, Tree-as-a-Prompt (TAP), aimed at bolstering both the performance and interpretability of LLMs when applied to tabular data. We have empirically demonstrated that incorporating tree-structured prompts with the original query significantly enhances the predictive capabilities of LLMs on such data. Furthermore, we have innovatively devised a method for training decision trees utilizing the feedback from LLMs, thereby fostering a synergistic relationship between the two models. This approach sheds light on the potential of integrating decision trees as a valuable auxiliary to amplify the efficacy of LLMs. In the future, we will generalize our approach to support other formats of data.

Our methodology uniquely amalgamates elements of both prompt tuning and prompt engineering. While we optimize the tree structure based on the feedback received from the LLMs, akin to prompt tuning, the tree-structured prompts simultaneously serve as structured rules, aiding the LLM in reasoning like chain-of-thoughts prompting.

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

APPENDICES

# A  PROMPTING TEMPLATES

In our experiments, the following prompting template is employed. Text in blue font denotes the beginning or end of a block, red font signifies variables that vary across datasets, orange font represents system commands, and text in black font is the fixed input text used for prompting.

```
1  % block system_directive %
2  You are a good data analyst that can find patterns of given data. You
       will be given a list of rules that you must follow, and you should
       predict the output of the given data based on the rules and your
       observation. Please don't give explanations.
3  % endblock %
4
5  % block dataset_intro %
6  Here's your task. The data below is used to [dataset.target]
7
8  The data consists of the following features:
9   for feature in dataset.features
10      [feature.name, feature.description]
11  endfor
12
13 For each data, the result can be one of the following:
14  for label in dataset.labels
15      [label.name, label.description]
16  endfor
17 % endblock %
18
19
20 % block format_descriptions %
21 Here's how the data will be presented. For each line:
22      [features], <RESULT>
23 % endblock %
24
25 % block incontext_examples %
26 These are some known data points:
27  for example in examples
28      [example]
29  endfor
30 % endblock %
31
32 % block tree_rules %
33 The rules are as follows:
34  for rule in rules
35      [rule]
36  endfor
37 % endblock %
38
39 For data points that are not covered by any rule, you should predict the
       result based on your observation.
40
41 % block prediction_intro %
42 Please make prediction on the following [n_instances] lines containing "<
       RESULT>", filling "<RESULT>" with one of [labels], and reply each
       prediction in a new line. Make sure there are exactly [n_instances]
       lines.
43  for instance in instances
44      [instance.features], <RESULT>
45  endfor
46 % endblock %
```

Table 4: The information of datasets used in our experiments.

| Datasets | #instances | #features | Tasks |
|---|---|---|---|
| diabetes | 768 | 8 | disease diagnosis |
| car | 1,275 | 6 | car acceptability prediction |
| blood | 748 | 4 | blood donation prediction |
| abalone | 4,177 | 8 | age prediction |

Table 5: Application of our approach on different LLMs on diabetes.

| Models | LLM | TAP-DT | TAP-OT |
|---|---|---|---|
| GPT-3.5-0301 | 0.78±0.02 | 0.78±0.02 | **0.79**±0.03 |
| Vicuna 7b-v1.5 | 0.55±0.02 | 0.57±0.01 | **0.60**±0.03 |

## B  STATISTICS OF DATASETS

The statistics of the datasets used in our experiments are presented in Table 4.

## C  DIFFERENT LLMS

For our primary experiments, we employ the June version of GPT-3.5. In this section, we assess the robustness of our approach across various LLMs. We evaluate two more models on the diabetes dataset in a single-shot setting: GPT-3.5-turbo-0301 (Legacy), and Vicuna 7b-v1.5 (Chiang et al., 2023). The respective results are tabulated in Table 5. The table reveals that although GPT-3.5 significantly outperforms Vicuna, our methodology consistently enhances the performance of each LLM, invariably surpassing the other baselines outlined in Table 1.

## D  TRAINING TIME

The training time of TAP-OT is presented in Table 6. The training is very fast when the number of training instances is small. When the number of training instances increases, the number of possible split values to traverse increases. Moreover, due to the rate limit of OpenAI API access, the inference time of LLM also increases. One technique to reduce the training time is to do the splitting for the nodes of the same level in parallel, which we will employ in the future.

Table 6: The training time of TAP-OT (s).

| #shots | diabetes | car | blood | abalone |
|---|---|---|---|---|
| 1 | 3.4 | 2.3 | 1.3 | 0.6 |
| 2 | 2.6 | 6.2 | 2.9 | 3.4 |
| 4 | 52.4 | 21.2 | 211.9 | 10.5 |
| 8 | 1000.5 | 168.1 | 938.7 | 488.4 |

