# OpenReview forum: "Tree-as-a-Prompt: Boosting Black-Box Large Language Models on Few-Shot Classification of Tabular Data"
_ICLR.cc/2024/Conference — ICLR 2024 Conference Withdrawn Submission_

### Official Review · Reviewer_3KZT · 2023-10-31

**Soundness:** 2 fair
**Presentation:** 4 excellent
**Contribution:** 2 fair
**Rating:** 3
**Confidence:** 4

**Summary:**

The authors propose a methodology to enhance the classification capabilities of LLMs on tabular data in a black-box setting using the few-shot paradigm. They show that appending trees as prompts into the LLMs improves the performance, and they also introduce a method to train a decision tree using the feedback from an LLM. The experiment shows superior performance over the LLM or the decision tree. Also, they show the possibilities of interpretation from those trees and the applicability of a federated learning paradigm.

**Strengths:**

Originality: the authors show a creative way to deal with tabular data in the context of LLMs.
Quality and Clarity: the paper is well-written and easy to follow. The methodology is well explained and capable of being reproduced.

**Weaknesses:**

Although the paper proposes a reasonable advance for the field of tabular classification in a few-shot manner, it would be good to have experiments on additional tabular datasets to better support the claims. Experiments related to the impact of the number of features will also be good. The proposed method can be applied to any LLM model, not only black-box ones; I do not see any good motive to focus solely on those. So, experiments using open-access LLMs are a need and will add valuable insights related to the method.

**Questions:**

How does the performance and time needed to train change regarding the increase in the number of features used?
Why not running experiments on additional datasets?

---

### Official Review · Reviewer_fSXp · 2023-11-01

**Soundness:** 2 fair
**Presentation:** 2 fair
**Contribution:** 1 poor
**Rating:** 3
**Confidence:** 3

**Summary:**

- The authors tackle the problem of applying LLMs to tabular data.
- The authors show that providing an LLM with information about a decision tree can improve performance.
- The authors present a method to train a decision tree using feedback from a large language model.
- Experiments are carried out vs. a number of baselines, across four datasets.

**Strengths:**

- (S0) This paper presents a novel idea in which we train a decision tree as an aid to a language model in answering queries pertaining to tabular data.

**Weaknesses:**

- (W0) While the paper has a neat idea, I'm not certain that the contribution and results are significant. There are some obvious baselines missing, the problem setting is quite niche, there are some obvious drawbacks of the method, and the writing is not very polished (see below). Overall the weaknesses below outweigh the strengths, and therefore I do not recommend acceptance to ICLR.

- (W1) The performance of TAP-OT is not very impressive: From Table 1, TAP-OT does not reliably beat baselines once the number of few-shot examples is 16 or more, and improvement margins (if present) are quite thin across the board.

- (W2) The paper tackles a problem that is quite niche. The usefulness of this method only arises in cases of very limited data such that a decision tree cannot be trained on its own.

- (W3) 1) I would like to see an LLM baseline using an optimized prompt (you don’t show this prompt). Since the datasets involved have few features and you are using very few training examples, it is hard to see why you need a complicated search procedure to provide the LLM this data. 2) I also would like to see a baseline that uses chain-of-thought and/or self-consistency (SC). I suspect you could improve the baseline LLM results by more than TAP-OT’s outperformance margin using just SC.

- (W4) The authors don’t show results for various choices of hyperparameter $\lambda$ and $\mu$. I presume the algorithm is not very robust to various choices of hyperparameters, and in the best case (after searching over a set of $\mu$s and $\lambda$s), you can marginally beat GPT3.5.

- (W5) TAP-OT requires an order of magnitude more calls to the LLM compared to the LLM baseline, before hyperparameter search over $\mu$ and $\lambda$. This is probably not worthwhile in most settings.

- (W6) The writing is not very polished. There are typos (e.g. “standard derivation”), and a few areas in which the claim is exaggerated (e.g. “our tree emerges as more rational”, “registering near-optimal accuracy”), and some sentences which are not clear (e.g. “...with scarce normal feature values” etc.). The explanation of the algorithm (Section 4) could also be clearer. Finally, related works on using LLMs to augment data should be mentioned. The diabetes dataset first mentioned isn't referenced until the 6th page.

**Questions:**

1) What is the prompt in the LLM baseline?

2) Can you clarify how many average LLM calls on line 15 in Algorithm 1 are there in each of your experiments in Table 1?

3) Could you confirm whether there is a different $\mu$ and $\lambda$ for each row in Table 1? (i.e. a different $\mu$ and $\lambda$ for each (dataset, #example) tuple).

4) Can you confirm how the OT baseline deals with training examples where the decision tree outputs "unknown"?

---

### Official Review · Reviewer_iRXR · 2023-11-01

**Soundness:** 3 good
**Presentation:** 4 excellent
**Contribution:** 3 good
**Rating:** 6
**Confidence:** 3

**Summary:**

In this paper, the authors propose an approach for tabular data representation using LLMs. They propose: (1) embedding feature-label relations for different datasets as a decision tree in addition to the text prompt, (2) training tree-based models in the loop with prompts.
Authors find that the use of these prompts improves performance on several standard UCI ML datasets.

**Strengths:**

1) Paper is clear and well-written and motivation/reasoning to propose the method is clear.
2) The greedy-tree training with LLMs is an interesting approach, and is generally a creative way to add domain knowledge in some manner to LLM prompting in an efficient manner
3) Results are interesting, and the authors perform a number of ablation studies including the use of federated learning and different tree structures

**Weaknesses:**

1) It is not clear why a tree specifically was chosen as the interpretable model. E.g., why not a linear classifier with regularization?
2) The interpretability analysis while interesting is based on a small number of examples, and thus may not be generalizable findings
3) The authors have proposed a method in some sense to incorporate domain knowledge (if the tree is appropriately trained) into the LLM's prompt: a more detailed literature review of prior work in this space would add to the paper

**Questions:**

1) Can authors clarify why a tree specifically was chosen as the interpretable model. E.g., why not a linear classifier with regularization?
2) Can authors expand on the interpretability analysis? E.g. note the average depth/max number of leaf nodes in each case

---

### Author Response · Authors · 2023-11-23

Dear Reviewers,

Thanks for your constructive comments! Due to the time constraint, we are not able to finish all the additional experiments you mentioned (e.g., experiments on more datasets, experiments on more LLMs). We will withdraw the paper and revise it in the future. Thanks again!

Regards,

Paper 770 Authors